# Leg Attachment Devices of Tiger Beetles (Coleoptera, Cicindelidae) and Their Relationship to Their Habitat Preferences

**DOI:** 10.3390/insects15090650

**Published:** 2024-08-29

**Authors:** Zheng Liu, Stanislav N. Gorb, Hongbin Liang, Ming Bai, Yuanyuan Lu

**Affiliations:** 1Key Laboratory of Animal Biodiversity Conservation and Integrated Pest Management, Institute of Zoology, Chinese Academy of Sciences, Beijing 100101, China; liuzheng@lfnu.edu.cn (Z.L.); lianghb@ioz.ac.cn (H.L.); baim@ioz.ac.cn (M.B.); 2Hebei Key Laboratory of Animal Diversity, Langfang Normal University, Langfang 065000, China; 3Department of Functional Morphology and Biomechanics, Institute of Zoology, Christian-Albrechts-University of Kiel, D-24118 Kiel, Germany; sgorb@zoologie.uni-kiel.de; 4University of Chinese Academy of Sciences, Beijing 100049, China

**Keywords:** microstructure, adhesive setae, tarsus, Cicindelidae, scanning electron microscopy

## Abstract

**Simple Summary:**

Adherence to smooth substrates is closely related to the morphology and distribution of adhesive structures on insects’ legs, so it is hypothesized that the adhesive structures have been evolved as an adaption to smooth substrates in specific environments. However, the factors that promote the evolution of adhesive structures are still unclear. Using scanning electron microscopy, we compared the microstructure of the tarsi of five tiger beetle species, both male and female, belonging to two tribes living in arboreal and non-arboreal environments. We found that the different types of adhesive setae, including elongated spoon-like setae, elliptical setae, branched setae, filament-like setae, discoidal setae, spatulate setae and tapered setae, varied in different environments and genders. The adaptive evolution of these adhesive structures was probably driven by the selective pressures of both mating behavior and the presence of smooth substrates in the respective environments.

**Abstract:**

The ability of many insects to adhere vertically or even upside down to smooth substrates is closely related to the morphology and distribution of the adhesive structures on their legs. During locomotion, the legs are in direct contact with different substrates, and it is hypothesized that the adhesive structures have been evolved as an adaption to smooth substrates in specific environments. To investigate whether there is a relationship between the presence of adhesive structures and the combined effects of different environments and mating behavior, we compared five species of tiger beetles belonging to two tribes living in arboreal and non-arboreal environments, respectively. In three non-arboreal species, we found a specific type of adhesive structure consisting of elongated spoon-like setae present on the protarsi of males but absent on the male meso- and metatarsi and on females. In *Tricondyla pulchripes*, an arboreal species living on stems, we found three types of adhesive setae on male protarsi, while only two types of setae were found on male meso- and metatarsi and on females. In *Neocollyris linearis*, an arboreal species living on leaves, we found three types of adhesive setae on male pro-, meso- and meta-tarsi but only two types of adhesive setae on females. The adaptive evolution of these adhesive structures was probably driven by the selective pressures of both mating behavior and the presence of smooth substrates in the respective environments. It is discussed that the adhesive structures in tiger beetles may be an adaptive evolutionary response to the plant surfaces and may play an important role in species differentiation.

## 1. Introduction

The morphological diversity and function of the adhesive structures have long been of interest to scientists [1,2,3,4]. Many insects have the ability to adhere to smooth surfaces, and their adhesive ability has been attributed to the attachment structures on their leg segments, such as tarsi, pretarsi or the distal end of tibia, which are often in direct contact with substrates during locomotion [5]. For example, ants use a soft, smooth arolium between the claws for attachment [6,7,8], flies have two hairy pulvilli under the claws [9,10,11,12,13], and leaf beetles rely on a large area of adhesive setae on the ventral side of the tarsomeres [14,15,16,17,18,19].

Morphological diversity in adhesive structures has been found in several insects [7,14,20,21,22,23,24], but the relationship between different adhesive microstructures and different environments has only been studied in a few selected species [4,25,26,27]. Generally, adhesive structures are more developed in the groups in close contact with plants [4,5,28,29,30]. We speculate that many phytophagous groups or some arboreal predatory groups may have experienced the selective pressure of smooth plant substrates in their environment, but more in-depth analyses and systematic studies are relatively scarce [31,32].

In the present paper, we explore the correlation between adhesive microstructures and specific environments, and we attempt to figure out the driving forces behind the emergence of adhesive structures in the adaptive evolutionary process of tiger beetles (Coleoptera: Adephaga: Cicindelidae), a group of predatory insects, which occupy diverse habitats (Figure 1). There are 130 genera and more than 2900 species of tiger beetles in the world’s fauna [33,34,35]. They are predators of small arthropods and are known for their extremely high running speed and highly developed visual system [36]. Based on the consensus of all modern phylogenomic studies, Duran and Gough recognized six tribes (Manticorini, Megacephalini, Collyridini, Ctenostomatini, Cicindelini and Oxycheilini) in the family Cicindelidae [37,38]. Amblycheilini has been recognized as an independent tribe in some studies [39], or as part of the tribe Megacephalini. Most species of the tribes Manticorini, Megacephalini, Cicindelini and Oxycheilini live on gravel ground and run fast, while other tribes, such as Collyridini and Ctenostomatini, are arboreal and live on trees, e.g., the genus *Neocollyris* [37]. The habitats of tiger beetles are more diverse than those of most families of Coleoptera.

The adhesive structures of tiger beetle legs have rarely been studied. Stork [14] examined one male tiger beetle (*Cicindela campestris*) by scanning electron microscopy (SEM) and found many adhesive setae on the three proximal tarsomeres of the protarsi. Pearson found hairy pads in some species [36], but the details of their microstructure have not been studied. Until now, there has been no comparative morphological study of the adhesive structures in more species, especially in tiger beetles occupying different habitats.

We used comparative morphological methods in our study. The tarsi of five tiger beetle species, living in two very different habitats (arboreal versus non-arboreal), were examined by SEM. It was hypothesized that the arboreal species have developed adhesive structures as an adaptation to the smooth leaf surfaces of plants. Since, in some beetle groups, the males use adhesive pads on their legs to attach themselves to the female during mating, we also compared the sexes of the species studied.

## 2. Materials and Methods

### 2.1. Materials

The adult specimens of five tiger beetle species (*Cicindela sachalinensis*, *Cosmodela separata*, *Cylindera kaleea*, *Tricondyla pulchripes* and *Neocollyris linearis*) were collected in China between 2008 and 2022. The specimens of *Ci. sachalinensis* were collected in Beijing, 2022. The specimens of *Co. separata* and *Cy. kaleea* were collected in Zhejiang Province, 2016. The specimens of *T. pulchripes* were collected in Hong Kong, 2017. The specimens of *N. linearis* were collected in Yunnan Province, 2008. All the specimens were from the National Animal Collection Resource Center of the Institute of Zoology, Chinese Academy of Sciences (IZCAS, Beijing, China). The specimens of *T. pulchripes* were dry preserved, whereas the others were preserved in 95% ethanol. Details of the studied specimens are provided in Table 1.

The five species were examined under a scanning electron microscope. The three non-arboreal species belong to tribe Cicindelini [36], whereas the two arboreal species belong to tribe Collyridini [37]. The tarsi (pro-, meso- and metatarsus) of one male and one female adult specimen of each species were examined and measured. The cuticle of the female elytra and ventral side of metasternum were examined.

### 2.2. Terminology

The morphological terminology mostly follows that of Stork (1980) [14], Beutel and Gorb (2001) [5] and Betz (2003) [40].

### 2.3. Photographic System

Observations were carried out under an Olympus SZ61 stereomicroscope. The digital images were taken with a Canon 5D digital camera in conjunction with a Canon MP-E 65 mm f/2.8 1-5X Macro Lens (Canon Inc., Tokyo, Japan). The digital camera flash system was composed of Nikon wireless remote speedlight SB-R200 (Nikon Inc., Tokyo, Japan) and Nikon wireless speedlight commander SU-800 (Nikon Inc., Tokyo, Japan). The Macro Lens was fitted to a StackShot macro rail (Cognisys Inc., Traverse, USA), then the images were stacked by Helicon Focus v.7.6.1. All the images were adjusted in Adobe Photoshop (Adobe Inc., San Jose, CA, USA).

### 2.4. Scanning Electron Microscopy (SEM)

The tarsi of the forelegs were removed from the body, cleaned with 2% phosphate buffered saline, stepwise dehydrated in ethanol (75%, 85%, 95%, 3 × 100%), CO_2_ critical-point-dried, coated with platinum, and then examined and photographed with a HITACHI SU8010 field emission scanning electron microscope (HITACHI Co. Ltd., Tokyo, Japan). The SEM images were post-processed with Adobe Photoshop (Adobe Inc.)

#### Morphometry of the Attachment System

The length and width of the setae were measured from the SEM images using Image-J 1.53 (National Institutes of Health, Bethesda, MD, USA) and displayed in the Results. To quantify the density of the setae on the tarsi, a 50 × 50 μm frame was applied on different areas of the SEM images using Image-J 1.53 (National Institutes of Health, USA). All the setae within the frame (50 × 50 μm = 2500 μm^2^) were counted and the calculated mean value (number of setae divided by 25) represents the density of setae per 100 μm^2^. Each part was measured three times and averaged (*n* = 3).

## 3. Results

The adhesive setae of males and females from all five species studied were different and sexually dimorphic (see the details below).

### 3.1. Cicindelini

*Cicindela sachalinensis* and *Cosmodela separata* are non-arboreal species rushing rapidly on the ground, and *Cylindera kaleea* usually lives on the ground or in low shrubs. Only males of these species have developed adhesive setae on the protarsi.

There are five tarsomeres in each tarsus. In males, the first to third tarsomeres of the protarsus are enlarged and widened (Figure 2A,G,M). They are covered with elongated spoon-like adhesive setae on the ventral side (Figure 2B,H,N). The fourth and fifth tarsomeres are nearly bald. The male meso- and metatarsi are without adhesive setae. The tarsomeres are slender, with thick and short setae on the ventral side (Figure 2C,D,I,O,P). In females, the tarsomeres are slender, without adhesive setae on all five tarsomeres of the pro-, meso- and metatarsi (Figure 2E,F,J,L,Q,R). A few short and thick setae are present on each tarsomere.

#### 3.1.1. *Cicindela sachalinensis* Morawitz, 1862

The fourth and fifth tarsomeres of males are nearly bald, with five pairs of short setae on the fourth tarsomeres and three on the fifth tarsomeres.

Elongated spoon-like setae (Figure 2B). Elongated spoon-like setae are the only type of seta on the ventral surface of the male protarsi. These setae were found on the first, second and third tarsomeres. The setal shaft is straight, about 89.50 ± 2.80 μm (*n* = 3) in length and 5.14 ± 0.33 μm (*n* = 3) in width (at the base). Each seta has an elongated spoon-like terminal plate at the tip. The ventral side of the terminal plate is regularly transversely ribbed, approximately 33.43 ± 2.77 μm (*n* = 3) in length and 8.19 ± 0.29 μm (*n* = 3) in width. The density of the elongated spoon-like setae is low: 0.32 setae per 100 μm^2^ (*n* = 3).

#### 3.1.2. *Cosmodela separata* (Fleutiaux, 1894)

The fourth and fifth tarsomeres of males are nearly bald, with five pairs of short setae on the fourth tarsomeres and three on the fifth tarsomeres.

Elongated spoon-like setae (Figure 2H). Elongated spoon-like setae are the only type of seta on the ventral surface of the male protarsi. These setae were found on the first, second and third tarsomeres. The setal shaft is straight, about 78.19 ± 6.71 μm (*n* = 3) in length and 4.97 ± 0.17 μm (*n* = 3) in width (at the base). Each seta has an elongated spoon-like terminal plate at the tip. The ventral side of the terminal plate is regularly transversely ribbed, approximately 43.66 ± 1.00 μm (*n* = 3) in length and 7.42 ± 0.30 μm (*n* = 3) in width. The density of the elongated spoon-like setae is rather low: 0.24 setae per 100 μm^2^ (*n* = 3).

#### 3.1.3. *Cylindera kaleea* (Bates, 1866)

The fourth and fifth tarsomeres of males are nearly bald, with five pairs of short and thick setae on the fourth and fifth tarsomeres separately.

Elongated spoon-like setae (Figure 2N). Elongated spoon-like setae are the only type of seta on the ventral surface of the male protarsi. These setae were found on the first, second and third tarsomeres. The setal shaft is straight, about 59.42 ± 3.69 μm (*n* = 3) in length and 5.57 ± 0.07 μm (*n* = 3) in width (at the base). Each seta has an elongated spoon-like terminal plate at the tip. The ventral surface of the terminal plate is regularly transversely ribbed, approximately 18.56 ± 1.90 μm (*n* = 3) in length and 7.65 ± 0.83 μm (*n* = 3) in width. The density of the elongated spoon-like setae is rather low: 0.24 setae per 100 μm^2^ (*n* = 3).

### 3.2. Collyridini

#### 3.2.1. Subtribe Tricondylina: *Tricondyla pulchripes* White, 1844

As an arboreal species living on tree stems, *T. pulchripes* has well-developed adhesive setae on all the tarsi of males and females, but the types of setae and their location are different in males and females.

In both genders, the general morphology of the tarsomeres is as follows. The first tarsomere (Tar I) is elongated. The second tarsomere is slightly enlarged and widened. The third and fourth tarsomeres are asymmetrical and larger on the lateral side (Figure. 3A,E,H,I,M). The ventral surfaces of the first, second, third and fourth tarsomeres are densely covered with adhesive setae. The fifth tarsomere (Tar V) is slender and covered with scattered thick and long setae on the ventral, lateral and dorsal surfaces (Figure 3J).

In males, there are three types of adhesive setae: elliptical setae, branched setae and filament-like setae. In females, there are two kinds of adhesive setae: branched setae and filament-like setae. The thick and long setae on the fifth tarsomere are presumably not adhesive.

Elliptical setae. These setae are situated on the ventral surfaces of the first, second and third tarsomeres of the male protarsi. The setal shaft is straight, about 54.61 ± 1.06 μm (*n* = 3) in length and 4.13 ± 0.12 μm (*n* = 3) in width (at the base). Each seta has an elliptical terminal plate at the end. The terminal plate is approximately 19.62 ± 0.88 μm (*n* = 3) in long diameter and 8.74 ± 0.43 μm (*n* = 3) in short diameter (Figure 3C). The elliptical setae have a relatively low density and there are only 0.48 setae per 100 μm^2^ (*n* = 3). There are transversal stripes on the ventral side of the setae.

Branched setae. These setae are situated on the ventral surface of the fourth tarsomere and on the surrounding edge of the first to third tarsomeres of the male protarsi (Figure 3D). Also, they are present on the ventral surfaces of the third and fourth tarsomeres of the male meso- and metatarsi (Figure 3F) and on the tarsi of all female legs (Figure 3K,O). These setae are elongate, slender, with small protuberances, about 148.12 ± 9.17 μm (*n* = 3) in length and 3.82 ± 0.04 μm (*n* = 3) in width (at the base). The apex of the seta is curved and acute. The average density of the branched setae is 1.68 setae per 100 μm^2^ (*n* = 3).

Filament-like setae. These setae are situated on the distal ventral surfaces of the first and second tarsomeres of all female legs and male meso- and metatarsi. On the setal shafts, these setae have grooves oriented at some angle to the setal axis. Their tips are gradually tapered (Figure 3G,L,P). The setae are about 90.17 ± 4.12 μm (*n* = 3) in length and 6.66 ± 0.41 μm (*n* = 3) in width (at the basis). The average density of the setae is 0.4 setae per 100 μm^2^ (*n* = 3).

#### 3.2.2. Subtribe Collyridina: *Neocollyris linearis* (Schmidt-Göbel, 1846)

*Neocollyris linearis* is an arboreal species living on trees. It has well-developed adhesive setae on the male and female tarsi, but the setal types and locations are different in males and females.

In both genders, the general morphology of tarsomeres is as follows. The first and second tarsomeres (Tar I and Tar II) are elongate. The third and fourth tarsomeres (Tar III and Tar IV) are slightly enlarged and wide (Figure 4A,E,G,I,O). The ventral surfaces of the first, second, third and fourth tarsomeres are densely covered with adhesive setae, which consist of two parts: setal shaft (sh) and specialized tip. The fifth tarsomere (Tar V) is wide and covered with scattered thick and short setae on the ventral surface (Figure 4D,J).

In males, there are three types of adhesive setae: discoidal, spatulate, and tapered setae. In females, there are only two types of adhesive setae: spatulate and tapered setae. The thick short setae on the fifth tarsomere are presumably not adhesive.

Discoidal setae. These setae are situated on the ventral surfaces of the first, second and third tarsomeres of the male tarsi (Figure 4C,H). The setal shaft is straight, with two or three grooves across the junction to the setal terminal plate, about 40.61 ± 1.15 μm (*n* = 3) in length and 3.56 ± 0.20 μm (*n* = 3) in width. Each seta has a discoidal terminal plate with 1–2 tips at the proximal part. The terminal plate is approximately 8.56 ± 0.54 μm (*n* = 3) in diameter (Figure 4C). The discoidal setae have a relatively low density: 0.6 setae per 100 μm^2^ (*n* = 3).

Spatulate setae. These setae are situated on the ventral surface of the fourth tarsomere in both male and female tarsi (Figure 4D,F,K,M,P). The setal shaft is straight, slightly bent, 64.71 ± 3.71 μm (*n* = 3) in length and 2.35 ± 0.14 μm (*n* = 3) in width. Each seta has a spatulate terminal plate. The widest part of the plate is 10.64 ± 0.48 μm (*n* = 3) in width (Figure 4D,K). The ventral and dorsal surfaces of the terminal plate are smooth, without obvious substructure. The spatulate setae have a high density: 1.8 setae per 100 μm^2^ (*n* = 3). These spatulate setae are very similar with those of leaf beetles [14,16,28] and ladybird beetles [14].

Tapered setae. These setae are situated on the ventral surfaces of the second and third tarsomeres of female legs (Figure 4L,N) and the surrounding edge of the first to third tarsomeres’ ventral surfaces of the male tarsi (Figure 4H). These setae are elongated, slender, about 66.42 ± 5.18 μm (*n* = 3) in length and 4.12 ± 0.09 μm (*n* = 3) in width (at the base). The setal shaft is straight at the base and tapered toward the curved acute tip. The density of the tapered setae is 1.12 setae per 100 μm^2^ (*n* = 3).

### 3.3. Microstructure of the Female Cuticle Surface and the Male Behavior during Mating

#### 3.3.1. Cicindelini

##### *Cicindela sachalinensis* and *Cosmodela separata*

During mating, the males of species that belong to this tribe always use the developed mandible to clasp the basic area of the pterothroax of the females (Figure 5A,G–I). The front legs do not always contact with the female elytra or her body side (Figure 5A,G).

By observing the cuticle surface of the female elytra and metasternum using SEM, it was found that there are regularly arranged hexagonal surface structures on the female elytra, with the concave surface in the hexagons (Figure 6A–C). The cuticle of the metasternum also has a hexagonal surface pattern, not concave, but rather a sheet-like with a lower height (Figure 6D–F). The surface of female *Cosmodela separata* has some roughness at the micro-scale. The width of the hexagons of *Ci. sachalinensis* is 15.63 ± 0.49 μm (*n* = 3) in the elytra and 14.98 ± 0.29 μm (*n* = 3) in the metasternum, respectively. The width of the hexagons of *Co. separata* is 13.47 ± 1.29 μm (*n* = 3) in the elytra and 12.56 ± 0.19 μm (*n* = 3) in the metasternum, respectively. Both *Ci*. *sachalinensis* and *Co*. *separata* have some kind of roughness on the elytra and *Co*. *separata* also some roughness on the metasternum.

#### 3.3.2. Collyridini (Subtribe Tricondylina)

##### *Tricondyla* *pulchripes*

The male of this species belongs to this tribe, which normally lacks a developed mandible. This is why the male normally uses its forelegs to contact the female lateral and ventral sides of the metasternum during mating (Figure 5B,C).

In SEM, the cuticle surface of the female elytra and metasternum have a regularly arranged hexagonal structures on elytra, but different from those on the cuticle of the tribe Cicindelini. The hexagonal structure is not concave inside, but rather sheet-like shaped and stepped with a lower height (Figure 6G). The female metasternum is long and sheet-like, not covered with hexagonal pattern (Figure 6J). There are many microscopical pores on the surface. Compared to the tribe Cicindelini, the cuticle is smoother at the micro-scale. The width of the hexagons is 17.01 ± 1.25 μm (*n* = 3) on the elytra.

#### 3.3.3. Collyridini (Subtribe Collyridina)

##### *Neocollyris* *linearis*

The males of species belonging to this tribe normally do not have developed mandibles. This is why they normally use their forelegs to contact the female cuticle on the dorsal side of the body (elytra and legs) during mating (Figure 5D–F).

Observing the cuticle surface of the female elytra and metasternum using SEM, it was found that the female elytra surface has an undulating pattern, with a regularly arranged hexagonal structure (Figure 6H). The hexagons have a uniform height. The width of the hexagons is 14.58 ± 0.54 μm (*n* = 3) in the elytra and 10.57 ± 0.77 μm (*n* = 3) in the metasternum, respectively. There are many micropores on the surface of the female elytra (Figure 6I) and metasternum (Figure 6L). The surface microstructure is similar on both these surfaces. The cuticle surface is smoother at the micro-scale than in *Co. separata* and is more similar to that of *T. pulchripes*.

## 4. Discussion

### 4.1. Correlation between Morphology of Adhesive Setae, Beetle Habitat and Gender

The tiger beetles in this study can be grouped into six categories according to their environments, where they live, and their gender (Figure 7): (1) female + non-arboreal; (2) male + non-arboreal; (3) female + arboreal stems; (4) male + arboreal stems; (5) female + arboreal leaves; (6) male + arboreal leaves. The different types of adhesive setae, including elongated spoon-like setae, elliptical setae, branched setae, filament-like setae, discoidal setae, spatulate setae and tapered setae, vary in different environments and genders.

In the female + non-arboreal type, there were two lines of thick setae on the ventral surface of the tarsus, but these setae were so large that they might not be helpful for adhesion on smooth substrates [18,28]. However, they may provide attachment through the mechanical interlocking mechanism [41]. In the male + non-arboreal type, there was only one type of adhesive setae, namely elongated spoon-like setae with an enlarged terminal end. Furthermore, this type of setae shows sexual dimorphism. We speculate that the sexual dimorphism in the adhesive setae may help males to attach to the female body surface. During mating, the legs of the adult males contact the body of the females. Since the female cuticle surface has an undulating pattern and is macroscopically uneven, achieving adhesion with tarsal structures difficult, so the male helps himself by clasping the basal area of the female pterothroax with his mandibles during mating.

In the female + arboreal stems type, there were two types of adhesive setae: filament-like setae and branched setae. In the male + arboreal stems type, the three types of adhesive setae were filament-like setae, branched setae, and elliptical setae. The elliptical setae are presumably “sexual setae” that play an important role in mating. The elytra of females and their metasternum surface are also smoother at the micro-scale, facilitating attachment by the tarsal setae. The filament-like setae and branched setae may be an adaptation to the arboreal environment. They are filamentous and may enhance mechanical interlocking on the rough stems. A similar shape of setae is known from the friction pads of chameleons [42], which are able to generate strong friction on rough plant stems.

In the female + arboreal leaves type, there were two types of adhesive setae: tapered and spatulate ones. In the male + arboreal leaves type, there were three types of adhesive setae: tapered, spatulate and discoidal ones. The spatulate terminal plate has an enlarged contact area to increase the adhesion force on various substrates, but especially on smooth substrates [16,43]. The discoidal seta in males is a different type of “sexual seta” also known from leaf beetles and ladybird beetles [14,16,28]. Compared with the elongated spoon-like and elliptical setae, they provide a larger contact area on smooth surfaces and with their specific shape generate more adhesion force. The surfaces of the female elytra and metasternum are rather smooth at the micro-scale, which facilitates attachment.

### 4.2. Comparison of Adhesive Devices within Tiger Beetles

In this study, the tarsal adhesive devices of two arboreal Cicindelini species and three non-arboreal Collyridini species were compared morphologically to investigate possible adaptations to different natural surfaces in their habitats. In general, the five tiger beetle species all have five tarsomeres in each tarsus and a pair of monodentate claws. The main differences between them lie in the structure and distribution of the adhesive setae on the ventral surface of the tarsomeres. The three non-arboreal species are very similar, with females lacking adhesive setae and males having elongated spoon-like setae on Tar I–III of the protarsi. However, the two arboreal species studied are distinct from each other. *Tricondyla pulchripes* has unique branched setae with many small filamentous protuberances not previously found in other groups of Cicindelidae [14,36]. The tapered, spatulate and discoidal setae of *Neocollyris linearis* are very similar to the adhesive setae of many leaf beetles (Chrysomelidae) and ladybird beetles (Coccinellidae), which are optimized for adhering to the rather smooth leaf surfaces of plants [14,16,17,19,28].

Recent molecular phylogenetic studies have recognized six tribes of tiger beetles (Manticorini, Megacephalini, Collyridini, Ctenostomatini, Cicindelini and Oxycheilini) [37,38]. Members of the tribes Collyridini and Ctenostomatini live in arboreal habitats and have an extremely slender body shape as an adaptation to this type of habitat [37]. In this study, *Neocollyris linearis* and *Tricondyla pulchripes* of the tribe Collyridini were shown to have hairy adhesive devices, confirming the adaptation of these species to arboreal habitats. The species of Cicindelini, Megacephalini and Manticorini are usually considered to be non-arboreal. The three species belonging to Cicindelini in this study lack adhesive setae, confirming their adaptation to non-arboreal habitats. Furthermore, the developed mandibles used by males during mating reduce the functional need for sexual adhesive setae.

### 4.3. Comparison of Adhesive Devices between Tiger Beetles and Other Beetles

Many groups of Coleoptera generally have hairy adhesive tarsomeres [5], especially species belonging to the suborder Polyphaga: they can climb well on rather smooth plant surfaces and bear adhesive structures, e.g., Chrysomelidae, Coccinellidae, Curculionidae, Cantharidae, Cerambycidae, and Staphylinidae [3,14,40,44,45,46]. However, not many studies have been carried out on representatives of Adephaga. Many Adephaga species live on the ground and the adhesive structures are rarely known for them. Previous research has shown that some male ground beetles (Carabidae) have sexually dimorphic setae on their tarsi [14]. Here, our results show that tiger beetles also have strongly developed adhesive setae, which in some species even resemble those of leaf beetles. The phylogenetic relationships of tiger beetles (Coleoptera: Adephaga: Cicindelidae) and leaf beetles (Coleoptera: Polyphaga: Chrysomelidae) are relatively distant, and they are even further distant from flies, spiders [47,48,49] or geckos [50], but they all have hairy adhesive structures adapted to smooth substrates in arboreal habitats. The adhesive devices are likely to be adaptive characters resulting from convergent evolution [4].

### 4.4. Key Drivers of the Specific Structure of Adhesive Setae

The tarsus is in direct contact with various surfaces in the environment during locomotion or grasping during mating. To adapt to the smooth surfaces, many insects have evolved microscopic adhesive setae on the ventral surface of their tarsi. Some male beetles use their claws to grasp the female’s body and the tarsus to make contact and adhere to the female’s surface during mating. This phenomenon was found in Chrysomelidae, Coccinellidae, Dytiscidae and the arboreal tiger beetle species *N. linearis* (this study). In non-arboreal tiger beetles, the male usually uses the developed mandible to clasp the base of the female’s pterothroax while its legs contact the female’s metasternum (this study). The male’s adhesive setae usually have round or elliptical terminal plates in Chrysomelidae, Coccinellidae and arboreal Cicindelidae, rectangular terminal plates in Carabinae, elongated spoon-like terminal plates in non-arboreal Cicindelidae (this study), or even true suckers in Dytiscidae [14,51], and they usually provide strong adhesion forces. Mating behavior is an important factor driving the evolution of adhesive microstructures and sexually dimorphic setae in male beetles.

The smooth substrates in the animal’s environment exert a strong selective pressure on both male and female tiger beetles. As the environments differ in the three habitats (on gravel grounds, on rough tree stems and on smooth tree leaves), the non-sexual adhesive setae exhibit different characters: without any adhesive setae, with dense rough branched setae, and with expanded terminal setae (spatulate), respectively. Under the double selective pressures of mating and smooth substrates in their specific habitats, the most developed adhesive structures are present in male arboreal species living on plant leaves.

It is known that the adhesive structures are more developed in those groups of animals that are closely associated with the plants they inhabit [4,5,25]. In order to adapt to the smooth surface of their host plants, adhesive structures evolve in the direction of increasing the effective contact area to enhance attachment performance. With the assistance of adhesive structures, insects extend their range of habitats and niches from rough surfaces to smooth leaves. It is reasonable for us to speculate that the emergence of adhesive structures in insects may be an adaptive evolutionary process to the widespread plants and may play an important role in species differentiation [26].

## 5. Conclusions

This study considers for the first time the effects of the gender and environment together as the common drivers of the evolution of leg adhesive structures in tiger beetles. We found seven types of adhesive setae and discussed the correlation between the female body surface, typical substrate surfaces in the beetle habitats and the morphology of the leg adhesive setae. This work provides a better understanding of the adaptive evolution of tiger beetles.

## Figures and Tables

**Figure 1 insects-15-00650-f001:**
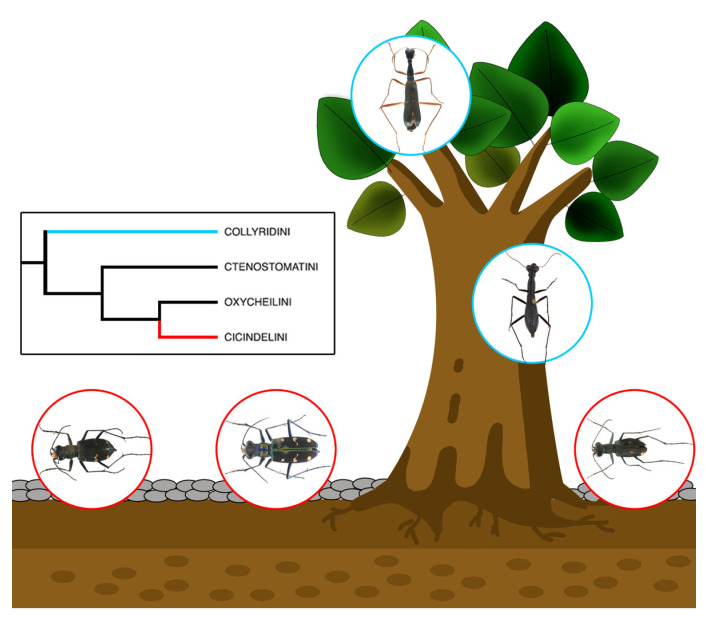
Arboreal and non-arboreal tiger beetle groups inhabiting different environments.

**Figure 2 insects-15-00650-f002:**
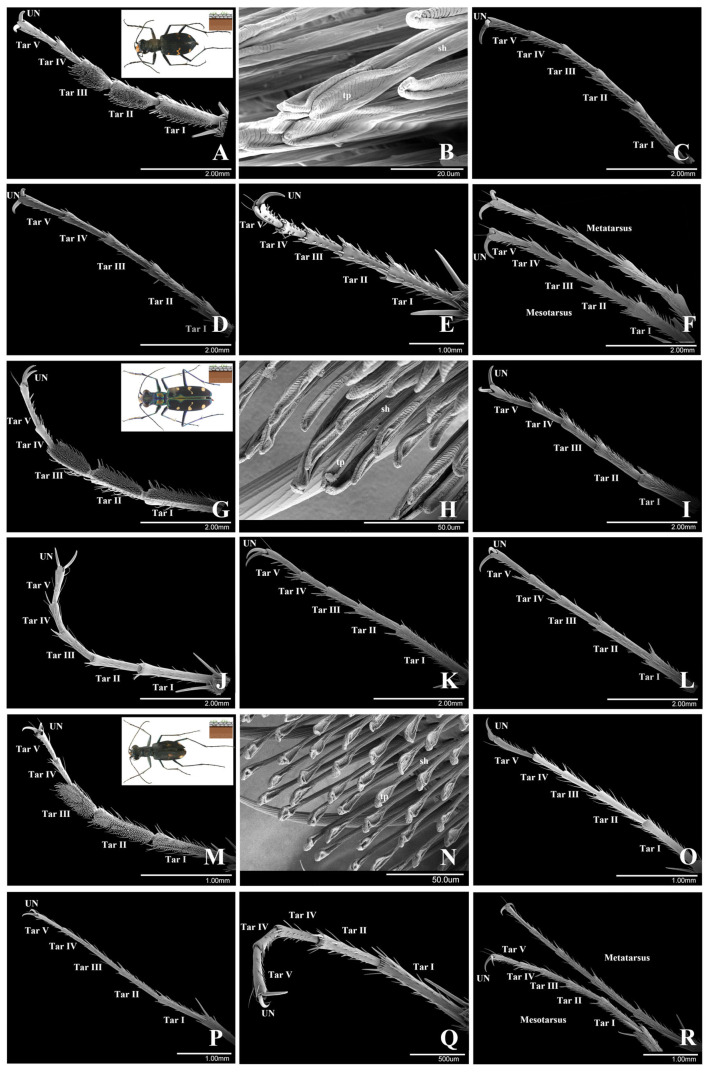
The ventral view of the tarsi and adhesive setae in non-arboreal species of Cicindelidae. (**A**–**F**). *Cicindela sachalinensis*. (**A**). Male protarsus. (**B**). Elongated spoon-like setae on the male protarsus. (**C**). Male mesotarsus. (**D**). Male metatarsus. (**E**). Female protarsus. (**F**). Female meso- and metatarsus. (**G**–**L**). *Cosmodela separata*. (**G**). Male protarsus. (**H**). Elongated spoon-like setae on the male protarsus. (**I**). Male mesotarsus. (**J**). Female protarsus. (**K**). Female mesotarsus. (**L**). Female metatarsus. (**M**–**R**). *Cylindera kaleea*. (**M**). Male protarsus. (**N**). Elongated spoon-like setae on the male protarsus. (**O**). Male mesotarsus. (**P**). Male metatarsus. (**Q**). Female protarsus. R. Female meso- and metatarsus. Abbreviations: UN, unguis (claw); Tar I, the 1st tarsomere; Tar II, the 2nd tarsomere; Tar III, the 3rd tarsomere; Tar IV, the 4th tarsomere; Tar V, the 5th tarsomere; sh, setal shaft; tp, terminal plate.

**Figure 3 insects-15-00650-f003:**
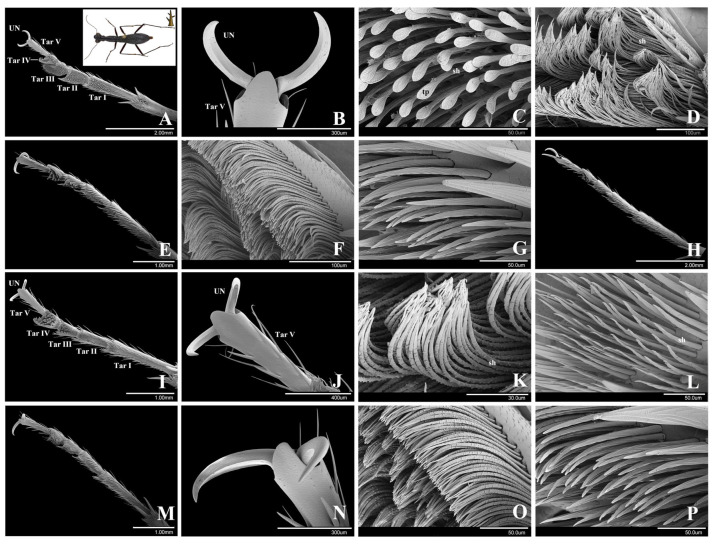
The tarsi and adhesive setae of *Tricondyla pulchripes* (arboreal species). (**A**–**H**). Male. (**A**). Protarsus, ventral view. (**B**). Unguis (claw) of protarsus. (**C**). Elliptical setae on protarsus. (**D**). Branched setae on protarsus. (**E**). Mesotarsus, ventral view. (**F**). Branched setae on mesotarsus. (**G**). Filament-like setae on mesotarsus. (**H**). Metatarsus, ventral view. (**I**–**P**). Female. (**I**). Protarsus, ventral view. (**J**). Unguis (claw) of protarsus. (**K**). Branched setae on protarsus. (**L**). Filament-like setae on protarsus. (**M**). Metatarsus, ventral view. (**N**). Unguis (claw) of metatarsus. (**O**). Branched setae on metatarsus. (**P**). Filament-like setae on metatarsus. Abbreviations: UN, unguis (claw); Tar I, the 1st tarsomere; Tar II, the 2nd tarsomere; Tar III, the 3rd tarsomere; Tar IV, the 4th tarsomere; Tar V, the 5th tarsomere; sh, setal shaft; tp, terminal plate.

**Figure 4 insects-15-00650-f004:**
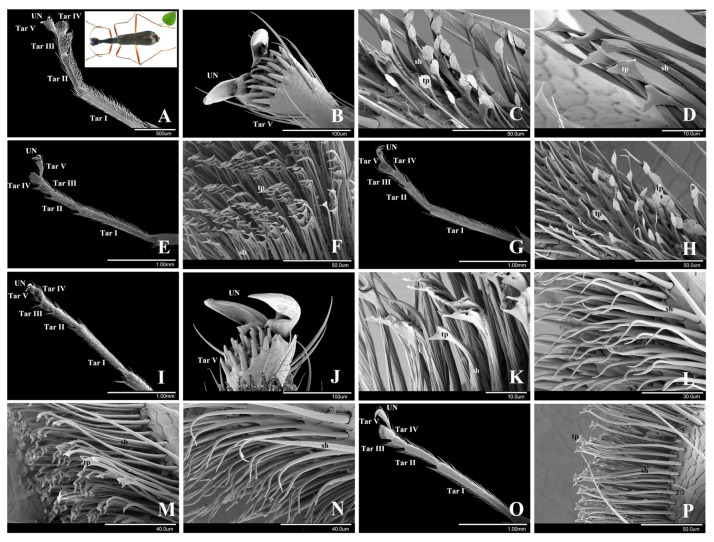
The tarsi and adhesive setae of *Neocollyris linearis* (arboreal species). (**A**–**H**). Male. (**A**). Protarsus, ventral view. (**B**). Unguis (claw) of protarsus. (**C**). Discoidal setae on protarsus. (**D**). Spatulate setae on protarsus. (**E**). Mesotarsus, ventral view. (**F**). Spatulate setae on mesotarsus. (**G**). Metatarsus, ventral view. (**H**). Discoidal setae on metatarsus. (**I**–**P**). Female. (**I**). Protarsus, ventral view. (**J**). Unguis (claw) of protarsus. (**K**). Spatulate setae on protarsus. (**L**). Tapered setae on protarsus. (**M**). Spatulate setae on mesotarsus. (N). Tapered setae on mesotarsus. (**O**). Metatarsus, ventral view. (**P**). Spatulate setae on metatarsus. Abbreviations: UN, unguis (claw); Tar I, the 1st tarsomere; Tar II, the 2nd tarsomere; Tar III, the 3rd tarsomere; Tar IV, the 4th tarsomere; Tar V, the 5th tarsomere; sh, setal shaft; tp, terminal plate.

**Figure 5 insects-15-00650-f005:**
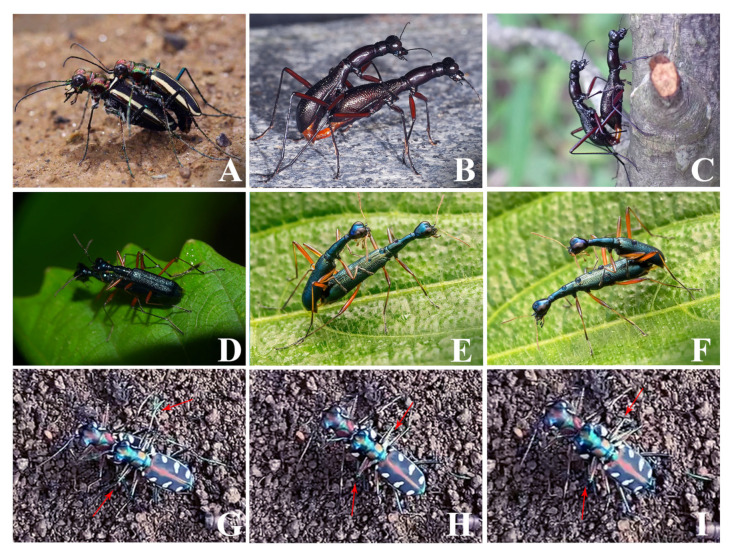
The contact position of the legs during mating. (**A**). Mating of *Ropaloteres desgodinsii* (Fairmaire, 1887) (Sichuan, photographed by Li He). (**B**). Mating of *Tricondyla pulchripes* (Hong Kong, photographed by Siuyeung Ho). (**C**). Mating of *Tricondyla pulchripes* (Hong Kong, photographed by Alfred Cheung). (**D**). Mating of *Neocollyris parvula* (Chaudoir, 1848) (Nagla Block, Palghar, Maharashtra, India, photographed by Dinesh Sharma). (**E**,**F**). Mating of *Neocollyris* sp. (Yeoor Hills, Thane West, Thane, Maharashtra, India, photographed by Anil Kumar Verma). (**G**–**I**). Screenshot from a video of the mating process of *Cosmodela juxtata* (Acciavatti and Pearson, 1989). (**G**). At 15 s (red arrows show that the protarsi of the male does not contact the body of the female). (**H**). At 20 s (red arrows show the protarsi of the male trying to contact the body of the female, and the male clasps the female with his mandibles). (**I**). At 22 s (red arrows show the same as in (**H**), the protarsi of the male contact with the body of the female more tightly).

**Figure 6 insects-15-00650-f006:**
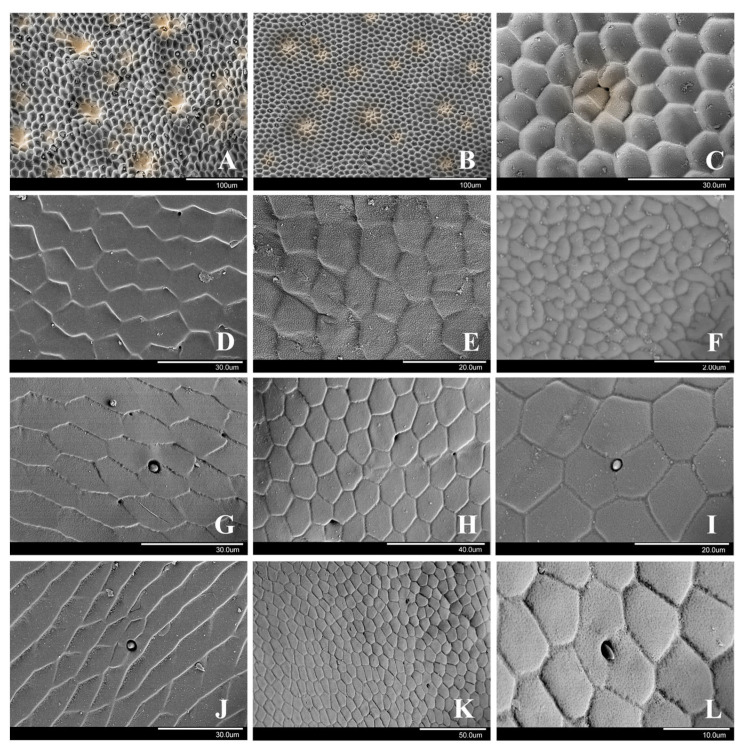
Microstructure of the elytra and metasternum in female tiger beetles. (**A**). Elytra of *Ci. sachalinensis*. (**B**). Elytra of *Co. separata*. (**C**). Elytra of *Co. separata*. (**D**). Metasternum of *Ci. sachalinensis*. (**E**). Metasternum of *Co. separata*. (**F**). Metasternum of *Co. separata*. (**G**). Elytra of *T. pulchripes*. (**H**). Elytra of *N. linearis*. (**I**). Elytra of *N. linearis*. (**J**). Metasternum of *T. pulchripes*. (**K**). Metasternum of *N. linearis*. (**L**). Metasternum of *N. linearis*. The yellowish regions in (**A**–**C**) indicate the raised parts.

**Figure 7 insects-15-00650-f007:**
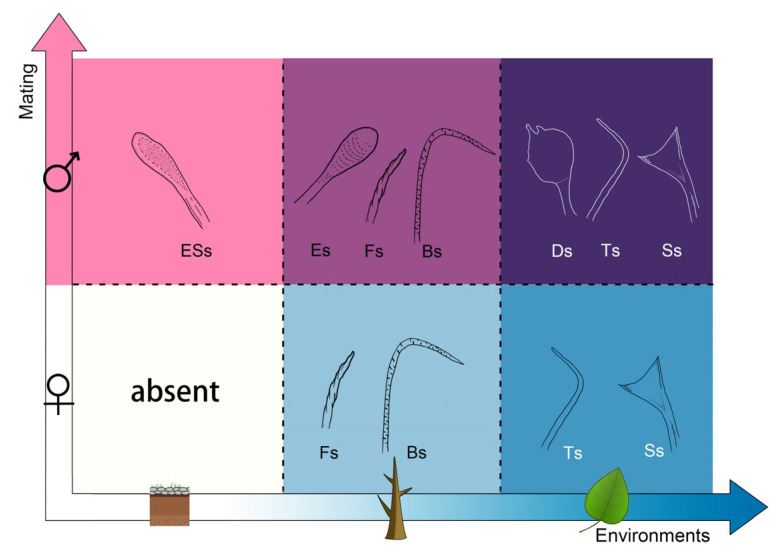
Scheme of possible correlations between the adhesive structures of the legs in both genders and the different environments where they live. The vertical axis represents the two genders: female (white) and male (pink), and the horizontal axis represents the smoothness of the habitat environments: ground (white), tree stem (light blue), and leaves (blue). ESs, elongated spoon-like setae; Es, elliptical setae; Fs, filament-like setae; Bs, branched setae; Ds, discoidal setae; Ts, tapered setae; Ss, spatulate setae.

**Table 1 insects-15-00650-t001:** Species of Cicindelidae and their collection locations.

	Tribe	Species	Collection Site	Habitat
1	Cicindelini	*Cicindela sachalinensis*	Wuling Mountain, Beijing	Non-arboreal
2	Cicindelini	*Cosmodela separata*	Jiangpu County, Zhejiang	Non-arboreal
3	Cicindelini	*Cylindera kaleea*	Siming Mountain, Zhejiang	Non-arboreal
4	Collyridini	*Tricondyla pulchripes*	Hong Kong	Arboreal, tree stem
5	Collyridini	*Neocollyris linearis*	Xishuangbanna, Yunnan	Arboreal, tree leaf

## Data Availability

The data presented in this study are available on request from the corresponding author.

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
