# Peer review of "Leg Attachment Devices of Tiger Beetles (Coleoptera, Cicindelidae) and Their Relationship to Their Habitat Preferences"

_insects, 2024, doi:10.3390/insects15090650_

Round 1

Reviewer 1 Report

Comments and Suggestions for Authors

The manuscript deals with the leg attachment devices of tiger beetles. The pictures are of good quality, but the manuscript is not well written. Therefore, I recommend to publish this paper in Insects after a major revision. In particular, the English language needs to be polished.

First of all, tiger beetles belong to the family Cicindelidae, not a subfamily of Carabidae, although there exist different opinions. Please refer to Duran & Gough (2020), “Validation of tiger beetles as distinct family (Coleoptera: Cicindelidae)”, Systematic Entomology, 45: 723-729.

Line 58: As a noun, the word research has no plural form.

Line 104: edited?

Lines 117-118: “number of setae divide 25” need to be rewritten.

Lines 192 and 235: “different in males and females” look strange. Please replace “in” with “from”.

Lines 336-337: “The adhesive pads of tiger beetles can be classified into six categories according to the environments, where they live, and gender”. Here the classifying standards (environments and gender, not proper words) need to be modified. In general, adhesive pads should be classified based on morphological features, I suggest.

Line 380: The term “prolegs” normally refer to hollow, paired, cylindrical outgrowths of the abdominal segments in Lepidoptera larvae. The same term cannot be used to refer to the forelegs here.

Comments on the Quality of English Language

The English language needs to be polished.

Author Response

1. Summary
Thank you very much for taking the time to review this manuscript. Your comments are very helpful. Please find the detailed responses below and the corresponding revisions highlighted in the re-submitted files. 

3. Point-by-point response to Comments and Suggestions for Authors
Comments 1: [The manuscript deals with the leg attachment devices of tiger beetles. The pictures are of good quality, but the manuscript is not well written. Therefore, I recommend to publish this paper in Insects after a major revision. In particular, the English language needs to be polished.]
Response 1: 
Thank you for pointing this out. We agree with this comment. Therefore, we have carefully polished the whole article.

Comments 2: [First of all, tiger beetles belong to the family Cicindelidae, not a subfamily of Carabidae, although there exist different opinions. Please refer to Duran & Gough (2020), “Validation of tiger beetles as distinct family (Coleoptera: Cicindelidae)”, Systematic Entomology, 45: 723-729.
Response 2: [(page 1, Lines 2-3) Leg Attachment Devices of Tiger Beetles (Coleoptera, Cicindelidae);
 (page 1, Line 41 Keywords) Cicindelidae;
 (page 2, Line 62) in the adaptive evolutionary process of tiger beetles (Coleoptera: Adephaga: Cicindelidae); 
(page 2, Line 66, 67) in the family Cicindelidae
(page 4, Line 104) Species of Cicindelidae and their collecting locations.
(page 6, Line 175) Figure 2. The ventral view of tarsi and adhesive setae in non-arboreal species of Cicindelidae.
(page 14, Line 385) not previously found in other groups of Cicindelidae
(page 14, Line 409) The phylogenetic relationships of tiger beetles (Coleoptera: Adephaga: Cicindelidae)
(page 14, Line 424) Coccinellidae and arboreal Cicindelidae
(page 14, Line 425) in non-arboreal Cicindelidae] 
Thank you for pointing this out. We entirely agree with this comment. Therefore, we have changed all the word “Cicindelinae” to “Cicindelidae” in this paper. 

Comments 3: [Line 58: As a noun, the word research has no plural form.]
Response 3: [(page 2, Line 67-68) in some studies.] Thank you for pointing this out. We agree with this comment. Another reviewer let us change 'researches' to 'studies', therefore, we have changed ' in some researches' to ' in some studies'.

Comments 4: [Line 104: edited?]
Response 4: [(page 4, Line 114) All images were adjusted in Adobe Photoshop (Adobe Inc.).] Thank you for pointing this out. The background of the image is not clean, so we use Photoshop to create a black background. I have not edited the section of tiger beetles. Therefore, we have deleted the word 'edited' and change the sentence to 'All images were adjusted in Adobe Photoshop (Adobe Inc.). '

Comments 5: [Lines 117-118: “number of setae divide 25” need to be rewritten.]
Response 5: [ (page 4, Line 127) number of setae divided by 25] Thank you for pointing this out. We fully agree with this comment. therefore, we have changed “divide” to “divided by”.

Comments 6: [Lines 192 and 235: “different in males and females” look strange. Please replace “in” with “from”.]
Response 6: [(page 7, line 188 and page 8, line 231) are different from males and females] Thank you for pointing this out. We agree with this comment. therefore, we have changed “different in males and females” to “different from males and females”.

Comments 7: [Lines 336-337: “The adhesive pads of tiger beetles can be classified into six categories according to the environments, where they live, and gender”. Here the classifying standards (environments and gender, not proper words) need to be modified. In general, adhesive pads should be classified based on morphological features, I suggest.]
Response 7: [(page 12, Lines 333-338) The tiger beetles in this study can be classified into six categories according to the environments, where they live, and gender (Figure 7): (1) Fe-male+non-arboreal; (2) Male+non-arboreal; (3) Female+arboreal-stems; (4) Male+arboreal-stems; (5) Female+arboreal-leaves; (6) Male+arboreal-leaves. The different types of adhesive setae, including elongated spoon-like setae, elliptical setae, branched setae, filament-like setae, discoidal setae, spatulate setae and tapered setae, vary in different environments and genders.] Thank you for pointing this out. We agree with this comment. The previous statement was unclear. The tiger beetles studied can be divided into six categories based on the environments and gender. They have different types of adhesive setae, which are distinguished by their morphology. Therefore, we have changed “The adhesive pads of tiger beetles” to “The tiger beetles in this study”.

Comments 8: [Line 380: The term “prolegs” normally refer to hollow, paired, cylindrical outgrowths of the abdominal segments in Lepidoptera larvae. The same term cannot be used to refer to the forelegs here.]
Response 8: [(page 14, line 383) on Tar I – III of protarsi.] Thank you for pointing this out. We agree with this comment. Therefore, we have changed “prolegs” to “protarsi”.

4. Response to Comments on the Quality of English Language
Point 1: The English language needs to be polished.
Response 1: Thank you. We have carefully polished the whole article.

5. Additional clarifications
[During this period, our collaborators in the tiger beetle classification conducted further identification and discussion of the species, and modified the genus name of one species (from “Cicindela separata” to “Cosmodela separata”).]

Reviewer 2 Report

Comments and Suggestions for Authors

This is a generally well-written paper on an interesting and understudied topic – the morphological evolution of leg attachment devices in tiger beetles and their relation to natural history/ecology. The authors lay out a clear case for their conclusions and the figures are very good. There are a few grammatical errors or awkward wordings that could be changed (see below). However, the discussion of tiger beetle systematics is out of date and needs correcting (see below). Otherwise, this is a fine paper and should be published following minor revisions.

Despite differences in sequencing technology and taxon sampling, since 2017, all major phylogenomic studies have recovered tiger beetles as a monophyletic clade sister to Carabidae or Carabidae plus Trachypachidae (López-López & Vogler, 2017; Zhang et al., 2018; Gustafson et al., 2019; McKenna et al., 2019; Gough et al., 2020), never nested within Carabidae. This was reviewed in Duran et al. 2020 and as such, tiger beetles were recognized as a distinct family based on the overwhelming amount of data brought to bear on the issue. The most recent world catalogue of tiger beetles also follows the most updated treatment of tiger beetles as a separate family (Wiesner 2020). It does not seem defensible to recognize tiger beetles as a subfamily of Carabidae at this point.

As such, Lines 2-4 (the title) should be changed to:

“Leg Attachment Devices of Tiger Beetles (Coleoptera, Cicindelidae) and their Relationship to their Habitat Preferences”

And lines 51-52 should be changed to:

“…in the adaptive evolutionary process of tiger beetles (Coleoptera: Adephaga: Cicindelidae).”

There are also numerous places in the Discussion where Cicindelinae should be changed to Cicindelidae.

In Line 53, the authors state that there are “more than 2300 species of tiger beetles” but now it is approximately 2900 (Wiesner 2020) and this should be changed/updated as well.

Again, there is a lack of current knowledge about tiger beetle systematics here. Lines 55-57 state that it is “generally agreed that there are five tribes” and the reference was from 2016. Duran & Gough 2020 showed that based on the consensus of all modern phylogenomic studies (2017 onward) there are 6 tribes.  Please review this paper.

Other minor edits:

Line 58:

Instead of saying “in some researches” it should say “in some studies”

Line 124, 142, etc.

“Cicindela sachalinensis” should be italicized in Line 124.  This is true in other places as well.

Line 357:

Should say: “playing a significant role in mating” not “playing significant role in mating”

References:

Duran, D. P.; Gough, H. M. (2020) Validation of tiger beetles as distinct family (Coleoptera: Cicindelidae), review and reclassification 531 of tribal relationships. Systematic Entomology 45, 723–729.

Gough, H.M., Allen, J.M., Toussaint, E.F.A., Storer, C.G. & Kawahara, A.Y. (2020) Transcriptomics illuminate the phylogenetic backbone of tiger beetles. Biological Journal of the Linnean Society, 195(3), 740–751. https://doi.org/10.1093/biolinnean/blz195.

Gustafson, G.T., Baca, S.M., Alexander, A.M. & Short, A.E.Z. (2019) Phylogenomic analysis of the beetle suborder Adephaga with comparison of tailored and generalized ultraconserved element probe performance. Systematic Entomology. https://doi.org/10.1111/syen.12413.

López-López, A. & Vogler, A.P. (2017) The mitogenome phylogeny of Adephaga (Coleoptera). Molecular Phylogenetics and Evolution, 114, 166–174.

McKenna, D.D., Shin, S., Ahrens, D. et al. (2019) The evolution and genomic basis of beetle diversity. Proceedings of the National Academy of Sciences USA, 116(49), 24729–24 737.

Zhang, S., Che, L., Li, Y., Liang, D., Pang, H., Slipinski, A. & Zhang, P. (2018) Evolutionary history of Coleoptera revealed by extensive sampling of genes and species. Nature Communications, 9(1), 1–11. https://dx.doi.org/10.1038/s41467-017-02644-4.

Wiesner, J. (2020) Checklist of the Tiger Beetles of the World. Winterwork.

Comments on the Quality of English Language

The English is generally acceptable.  Only a few minor changes are suggested (see above).

Author Response

1. Summary        
Thank you very much for taking the time to review this manuscript. Your comments are very helpful. Please find the detailed responses below and the corresponding revisions highlighted in the re-submitted files. 

3. Point-by-point response to Comments and Suggestions for Authors
Comments 1: [This is a generally well-written paper on an interesting and understudied topic – the morphological evolution of leg attachment devices in tiger beetles and their relation to natural history/ecology. The authors lay out a clear case for their conclusions and the figures are very good. There are a few grammatical errors or awkward wordings that could be changed (see below). However, the discussion of tiger beetle systematics is out of date and needs correcting (see below). Otherwise, this is a fine paper and should be published following minor revisions.
Despite differences in sequencing technology and taxon sampling, since 2017, all major phylogenomic studies have recovered tiger beetles as a monophyletic clade sister to Carabidae or Carabidae plus Trachypachidae (López-López & Vogler, 2017; Zhang et al., 2018; Gustafson et al., 2019; McKenna et al., 2019; Gough et al., 2020), never nested within Carabidae. This was reviewed in Duran et al. 2020 and as such, tiger beetles were recognized as a distinct family based on the overwhelming amount of data brought to bear on the issue. The most recent world catalogue of tiger beetles also follows the most updated treatment of tiger beetles as a separate family (Wiesner 2020). It does not seem defensible to recognize tiger beetles as a subfamily of Carabidae at this point.
As such, Lines 2-4 (the title) should be changed to:
“Leg Attachment Devices of Tiger Beetles (Coleoptera, Cicindelidae) and their Relationship to their Habitat Preferences”
And lines 51-52 should be changed to:
“…in the adaptive evolutionary process of tiger beetles (Coleoptera: Adephaga: Cicindelidae).”
There are also numerous places in the Discussion where Cicindelinae should be changed to Cicindelidae.]

Response 1: 
[(page 1, Lines 2-3) Leg Attachment Devices of Tiger Beetles (Coleoptera, Cicindelidae);
 (page 1, Line 41 Keywords) Cicindelidae;
 (page 2, Line 62) in the adaptive evolutionary process of tiger beetles (Coleoptera: Adephaga: Cicindelidae); 
(page 2, Line 66, 67) in the family Cicindelidae
(page 4, Line 104) Species of Cicindelidae and their collecting locations.
(page 6, Line 175) Figure 2. The ventral view of tarsi and adhesive setae in non-arboreal species of Cicindelidae.
(page 14, Line 385) not previously found in other groups of Cicindelidae
(page 14, Line 409) The phylogenetic relationships of tiger beetles (Coleoptera: Adephaga: Cicindelidae)
(page 14, Line 424) Coccinellidae and arboreal Cicindelidae
(page 14, Line 425) in non-arboreal Cicindelidae] 
Thank you for pointing this out. We entirely agree with this comment. Therefore, we have changed all the word “Cicindelinae” to “Cicindelidae” in this paper. 

Comments 2: [In Line 53, the authors state that there are “more than 2300 species of tiger beetles” but now it is approximately 2900 (Wiesner 2020) and this should be changed/updated as well.]

Response 2: [(page 2, Line 62) There are 130 genera and more than 2900 species of tiger beetles in the world’s fauna [33–35].] Thank you for pointing this out. We agree with this comment. Therefore, we have changed “2300” to “2900”, and add this paper in the References.

Comments 3: [ Again, there is a lack of current knowledge about tiger beetle systematics here. Lines 55-57 state that it is “generally agreed that there are five tribes” and the reference was from 2016. Duran & Gough 2020 showed that based on the consensus of all modern phylogenomic studies (2017 onward) there are 6 tribes.  Please review this paper.]
Response 3: [(page 2, Line 64-67) Based on the consensus of all modern phylogenomic studies, Duran & Gough recognized six tribes (Manticorini, Megacephalini, Collyridini, Ctenostomatini, Cicindelini and Oxycheilini) in the family Cicindelidae [37, 38].] Thank you for pointing this out. We agree with this comment. Therefore, we change “five tribes” to “six tribes”, and add this paper in the References.

Comments 4: [Line 58: Instead of saying “in some researches” it should say “in some studies”]
Response 4: [(page 2, Line 67-68) in some studies]. Thank you for pointing this out. We agree with this comment. therefore, we change 'researches' to 'studies'.

Comments 5: [Line 124, 142, etc. “Cicindela sachalinensis” should be italicized in Line 124.  This is true in other places as well.]
Response 5: Thank you for pointing this out. We fully agree with this comment. therefore, we have changed all the species names to be italicized in this paper.

Comments 6: [Line 357: Should say: “playing a significant role in mating” not “playing significant role in mating”]
Response 6: [(page 13, line 360) play an important role in mating.] Thank you for pointing this out. We agree with this comment. therefore, we have changed “playing significant role in mating” to “play an important role in mating”.

4. Response to Comments on the Quality of English Language
Point 1: The English is generally acceptable.  Only a few minor changes are suggested (see above).
Response 1: Thank you. We have carefully polished the whole article.

5. Additional clarifications
[During this period, our collaborators in the tiger beetle classification conducted further identification and discussion of the species, and modified the genus name of one species (from “Cicindela separata” to “Cosmodela separata”).]

Reviewer 3 Report

Comments and Suggestions for Authors

LIU et al. manuscript for Insects: Specific comments Lines 125 to 177 include much redundancy. Since the structures of all three species are essentially the same, the tarsi and setae of all could be described together, and indicating any differences among these 3 species which seem minimal… Some examples of missing or poor wording, other comments: 118 divided by 145, In male 279 that belongs 336, believe you should say the tiger beetles in this study….. The caption for Figure 7 should include the 3 different environments shown and the types of setae illustrated in the figure…. 413// The tarsus 435 closely associated with the plants they inhabit

Comments on the Quality of English Language

see above

Author Response

1. Summary
Thank you very much for taking the time to review this manuscript. Your comments are very helpful. Please find the detailed responses below and the corresponding revisions highlighted in the re-submitted files. 

3. Point-by-point response to Comments and Suggestions for Authors

Comments 1: LIU et al. manuscript for Insects: Specific comments Lines 125 to 177 include much redundancy. Since the structures of all three species are essentially the same, the tarsi and setae of all could be described together, and indicating any differences among these 3 species which seem minimal…

Response 1: [page 5, lines 133-143].[Cicindela sachalinensis and Cosmodela separata are non-arboreal species rushing rapidly on the ground, and Cylindera kaleea usually lives on the ground or in low shrubs. Only males of them have developed adhesive setae on the protarsi.

There are five tarsomeres in each tarsus. In male, the 1st – 3rd tarsomeres of protarsus are enlarged and widened (Figure 2A, G, M). They are covered with elongated spoon-like adhesive setae on the ventral side (Figure 2B, H, N). The 4th and 5th tarsomeres are nearly bald. The male meso- and metatarsi are without adhesive setae. The tarsomeres are slender, with thick and short setae on the ventral side (Figure 2C, D, I, O, P). In female, the tarsomeres are slender, without adhesive setae on all the five tarsomeres of pro-, meso- and metatarsi (Figure 2E, F, J, L, Q, R). Few short and thick setae are present on each tarsomere.] Thank you for pointing this out. We fully agree with this comment. Therefore, we simplify the repetitive descriptive part. However, as the measurement data of the setae are different, we write the different parts separately.

Comments 2: other comments: 118 divided by 145, In male 279 that belongs 336, believe you should say the tiger beetles in this study…

Response 2: (page 4, Line 127) divided by; (page 5, Line 136) In male; (page 10, Line 275) that belongs;  (page 12, Line 333)The tiger beetles in this study].Thank you for pointing these out. We agree with these comments. We change “divide” to “divided by”; change “belongs” to “that belongs”; add “in this study” after “The tiger beetles”.

Comments 3: The caption for Figure 7 should include the 3 different environments shown and the types of setae illustrated in the figure….

Response 3: [(page 13, Lines 340-345) Figure 7. Scheme of possible correlations between adhesive structures of legs in both genders and different environments, where they live. The vertical axis represents two genders: female (white) and male (pink), and the horizontal axis represents the smoothness of the habitat environments: ground (white), tree stem (light blue), and leaves (blue). ESs, elongated spoon-like setae; Es, elliptical setae; Fs, filament-like setae; Bs, branched setae; Ds, discoidal setae; Ts, tapered setae; Ss, spatulate setae.] Thank you for pointing this out. We fully agree with this comment. Therefore, we add the 3 different environments and the types of setae in the caption.

Comments 4: 413, The tarsus 435 closely associated with the plants they inhabit

Response 4: [(page 14, Line 415)The tarsus; (page 15, Line 437) closely associated with the plants they inhabit] Thank you for pointing these out. We agree with these comments. We change “tarsus” to “The tarsus”; change “closely related to plants” to “closely associated with the plants they inhabit”.

4.Response to Comments on the Quality of English Language
Point 1:see above
Response1: Thank you. We have carefully polished the whole article.

5. Additional clarifications
[During this period, our collaborators in the tiger beetle classification conducted further identification and discussion of the species, and modified the genus name of one species (from “Cicindela separata” to “Cosmodela separata”).]

Round 2

Reviewer 1 Report

Comments and Suggestions for Authors

All the questions I addressed have been answered. I have no further questions now.